# GSTZ1-1 Deficiency Activates NRF2/IGF1R Axis in HCC via Accumulation of Oncometabolite Succinylacetone

Fan Yang[1,2,†], Jingjing Li[1,†], Haijun Deng[1,†], Yihao Wang[3], Chong Lei[1], Qiujie Wang[1], Jin Xiang[1], Li Liang[1], Jie Xia[1], Xuanming Pan[1], Xiaosong Li[2], Quanxin Long[1], Lei Chang[3], Ping Xu[3], Ailong Huang[1,*] , Kai Wang[1,**] & Ni Tang[1,***]

## Abstract

The IGF1R signaling is important in the malignant progression of cancer. However, overexpression of IGF1R has not been properly assessed in HCC. Here, we revealed that GSTZ1-1, the enzyme in phenylalanine/tyrosine catabolism, is downregulated in HCC, and its expression was negatively correlated with IGF1R. Mechanistically, GSTZ1-1 deficiency led to succinylacetone accumulation, alkylation modification of KEAP1, and NRF2 activation, thus promoting IGF1R transcription by recruiting SP1 to its promoter. Moreover, inhibition of IGF1R or NRF2 significantly inhibited tumor-promoting effects of *GSTZ1* knockout *in vivo*. These findings establish succinylacetone as an oncometabolite, and GSTZ1-1 as an important tumor suppressor by inhibiting NRF2/IGF1R axis in HCC. Targeting NRF2 or IGF1R may be a promising treatment approach for this subset HCC.

**Keywords**  GSTZ1-1; IGF1R; KEAP1; NRF2; succinylacetone
**Subject Categories**  Cancer; Metabolism
**The EMBO Journal (2019) 38: e101964**

## Introduction

The insulin-like growth factor (IGF) signaling axis is critical in the growth and development of various tissues. Upon binding to the IGF ligand, IGF-1 receptor (IGF1R) is activated through autophosphorylation. The two main downstream signals of IGF1R are the phosphatidylinositol-3 kinase (PI3K)/AKT and Ras/Raf/extracellular signal-regulated kinase (ERK) pathways, playing a prominent role in cellular proliferation, differentiation, and apoptosis (Yu & Rohan, 2000). Increasing evidence shows that IGF1R and its ligands (IGF-1, IGF-2) play an important role in the development and progression of several cancers, including hepatocellular carcinoma (HCC) (Aleem *et al*, 2011; Pollak, 2012). Although overexpression and activation of IGF1R have not been properly assessed in HCC, several approaches targeting IGF1R have been developed in an effort to engineer therapeutic strategies for HCC (Lodhia *et al*, 2015).

Metabolic reprogramming to support the bioenergetic and biosynthetic demands of rapid cell proliferation is a hallmark of cancer (Hanahan & Weinberg, 2011). Cancer cells exhibit enhanced glucose uptake coupled to lactate extrusion even in oxygen-rich conditions, which is well known as the Warburg effect (Ward & Thompson, 2012). In addition to altered glucose metabolism, changes in amino acid metabolism also contribute to different aspects of tumorigenesis; for example, aberrant activation of the serine/glycine biosynthetic pathway is critical in cancer pathogenesis (Amelio *et al*, 2014). Restoring these altered metabolic pathways may be a new cancer therapeutic approach (Vander Heiden, 2011).

Glutathione S-transferase zeta 1-1 (GSTZ1-1), also known as maleylacetoacetate isomerase (MAAI, EC 5.2.1.2), catalyzes the glutathione-dependent isomerization of maleylacetoacetate (MAA) to fumarylacetoacetate (FAA; Fernández-Cañón & Peñalva, 1998) and is the penultimate enzyme of phenylalanine/tyrosine (Phe/Tyr) catabolism. GSTZ1-1 is a member of the glutathione S-transferase (GST) superfamily, which catalyzes the reaction of glutathione with endo- and xenobiotics, and thus plays a critical role in cellular detoxification (Allocati *et al*, 2018). Further, GSTZ1-1 catalyzes the oxygenation of dichloroacetic acid (DCA) to glyoxylic acid and plays a role in xenobiotic α-haloacid metabolism (Tong *et al*, 1998). *Gstz1*$^{-/-}$ mice accumulated FAA and succinylacetone (SA) in the urine. When stressed with a high-Phe diet, these mice showed rapid

1   Key Laboratory of Molecular Biology for Infectious Diseases (Ministry of Education), Department of Infectious Diseases, The Second Affiliated Hospital, Institute for Viral Hepatitis, Chongqing Medical University, Chongqing, China
2   Department of Infectious Diseases, The First Affiliated Hospital, Chongqing Medical University, Chongqing, China
3   State Key Laboratory of Proteomics, Beijing Proteome Research Center, National Center for Protein Sciences (Beijing), Beijing Institute of Lifeomics, Beijing, China
    *Corresponding author. Tel: +86 23 68486780; Fax: +86 23 68486780; E-mail: ahuang@cqmu.edu.cn
    **Corresponding author. Tel: +86 23 68486780; Fax: +86 23 68486780; E-mail: wangkai@cqmu.edu.cn
    ***Corresponding author. Tel: +86 23 68486780; Fax: +86 23 68486780; E-mail: nitang@cqmu.edu.cn
    †These authors contributed equally to this work.

weight loss, renal and hepatic damage, necrosis, and lethality (Fernández-Cañón *et al*, 2002). Deficiency of GSTZ1-1 in mice causes constitutive oxidative stress and suggests the activation of the nuclear factor erythroid 2-related factor 2 (NRF2) antioxidant response pathway (Blackburn *et al*, 2006). Recently, GSTZ1-1 was reported to be downregulated in HCC and upregulated in breast cancer (Jahn *et al*, 2016), indicating that dysregulation of GSTZ1-1 may be involved in the tumorigenesis in humans. However, the underlying mechanism remains largely unknown.

NRF2, a master regulator of detoxification and the antioxidant response, is also involved in the regulation of metabolism and other essential cellular functions. It has been recognized as a driver of cancer progression, metastasis, and resistance to chemotherapy (Rojo de la Vega *et al*, 2018). The cysteine-rich protein kelch-like ECH-associated protein 1 (KEAP1), a central negative regulator of NRF2, is a substrate adaptor protein for cullin-3-containing E3 ubiquitin ligase and is responsible for NRF2 ubiquitylation and degradation. Under exposure to electrophiles or oxidative stress, several cysteine residues on KEAP1 are alkylated or oxidized, which results in a conformational change, enabling the dissociation of NRF2. NRF2 then translocates into the nucleus to induce expression of its target genes, such as *NAD(P)H:quinone oxidoreductase 1 (NQO1), heme oxygenase 1 (HO1),* and *glutamate-cysteine ligase catalytic/modifier subunit (GCLC/M),* through binding to the antioxidant responsive element (ARE) in the promoter regions of these genes (Nioi *et al*, 2003; Hayes *et al*, 2015).

In the present study, we found that the expression of GSTZ1-1 is downregulated in HCC cell lines and tumor tissues, and *Gstz1*$^{-/-}$ mice showed increased hepatocarcinogenesis following chemical carcinogen exposure. Moreover, we investigated whether GSTZ1-1 deficiency can upregulate IGF1R expression, activate IGF1R-mediated antiapoptotic pathway, and promote HCC progression. Here, we show that GSTZ1-1 serves as a tumor suppressor in HCC and provide an alternative mechanism by which an oncometabolite may activate the IGF1R pathway.

# Results

## GSTZ1-1 expression is downregulated in HCC and predicts poor prognosis of patients

To identify genes that are deregulated in HCC, we analyzed differential gene expression in paired HCC and normal para-carcinoma tissues in epithelial cell adhesion molecule (EpCAM)-positive and EpCAM-negative tumors, using GSE5975 profiles from the Gene Expression Omnibus database. EpCAM-positive and EpCAM-negative tumors shared 398 probes, corresponding to 348 genes that were differentially expressed between cancer and control tissues. Pathway analysis demonstrated that the differentially expressed genes were enriched in 26 pathways, including bile secretion, retinol metabolism, and tyrosine metabolism (Appendix Fig S1). One of the candidate genes (Table EV1), *GSTZ1* (belonging to tyrosine metabolism), was selected for further study.

We further analyzed *GSTZ1* mRNA expression in an independent cohort of 363 HCC tissues (including 50 paired tumor and normal liver tissues) from The Cancer Genome Atlas (TCGA) database. *GSTZ1* mRNA expression was significantly decreased in tumor tissues compared with normal liver tissues ($P < 0.001$; Fig 1A). Decreased GSTZ1-1 expression was associated with advanced tumor stage in the TCGA database (stage I/II versus stage III/IV, $P = 0.005$; Fig 1B). Kaplan–Meier's survival curve analysis demonstrated that patients with low expression of GSTZ1-1 had lower overall survival than patients with high expression of GSTZ1-1 (median survival, 1,372 versus 2,131 days, $P = 0.008$; Fig 1C).

Next, we examined GSTZ1-1 expression in 40 paired clinical HCC and normal tissue samples. Quantitative RT–PCR (qRT–PCR), Western blotting, and immunohistochemical (IHC) assay revealed that *GSTZ1* mRNA and protein expression levels were significantly lower in HCC than in the corresponding non-tumor tissues (Fig 1D–F). Collectively, these data indicated that downregulation of GSTZ1-1 in HCC may contribute to disease progression and predicts poor prognosis.

## GSTZ1-1 suppresses HCC cell proliferation *in vitro* and *in vivo*

To evaluate the role of GSTZ1-1 in HCC, we first checked endogenous GSTZ1-1 levels in a collection of hepatoma cell lines and MIHA cells (an immortalized human hepatocyte cell line). GSTZ1-1 protein expression was substantially lower in most hepatoma cells than in MIHA cells (Fig 2A). We constructed a recombinant adenovirus encoding *GSTZ1* (AdGSTZ1) to overexpress GSTZ1-1 (GSTZ1-OE), and an adenovirus expressing green fluorescent protein (AdGFP) was used as a control. In addition, we established *GSTZ1* knockout (GSTZ1-KO) HepG2 cell lines using the CRISPR/Cas9 system (Fig EV1). Overexpression and knockout efficiencies were confirmed by immunoblot assay (Fig 2B). GSTZ1-OE significantly repressed the proliferation of both Huh7 and SK-Hep1 cells, as shown by EdU incorporation, MTS, and colony formation assays, whereas GSTZ1-KO promoted HepG2 cell proliferation (Fig 2C–E).

To examine whether such effects exist *in vivo*, AdGSTZ1-, AdGFP-, or mock-infected MHCC-97H cells were injected subcutaneously into the flanks of nude mice. Xenograft tumors in the AdGSTZ1 group were significantly smaller than those in the AdGFP and mock control groups (Fig EV2A and B). Accordingly, xenograft weight was significantly lower in the AdGSTZ1 group ($P < 0.001$; Fig EV2C). Together, these results suggested that GSTZ1-1 can repress HCC cell proliferation both *in vitro* and *in vivo*.

## Loss of GSTZ1-1 results in succinylacetone accumulation and NRF2 activation

*Gstz1*$^{-/-}$ mice exhibit increased oxidative stress and activation of antioxidant response pathways (Blackburn *et al*, 2006). Thus, we evaluated the effect of GSTZ1-1 on the NRF2 activation. GSTZ1-OE decreased ARE luciferase activity, NRF2 nuclear translocation, and expression levels of NRF2 target genes, including *NQO1, multidrug resistance protein 2 (MRP2), GCLM, thioredoxin reductase 1 (TXNRD1), glucose-6-phosphate dehydrogenase (G6PD),* and *HO1* (Fig 3A–C top, and Fig EV3A). In contrast, GSTZ1-KO promoted activation of NRF2 antioxidant pathway (Fig 3A–C bottom). These data indicated that loss of GSTZ1-1 leads to NRF2 activation.

To gain a better understanding of the molecular mechanisms underlying NRF2 activation, *Gstz1*$^{-/-}$ mice on 129SvJ background were obtained from the European Mouse Mutant Archive. Consistent with previous studies (Fernández-Cañón *et al*, 2002), we

                                        

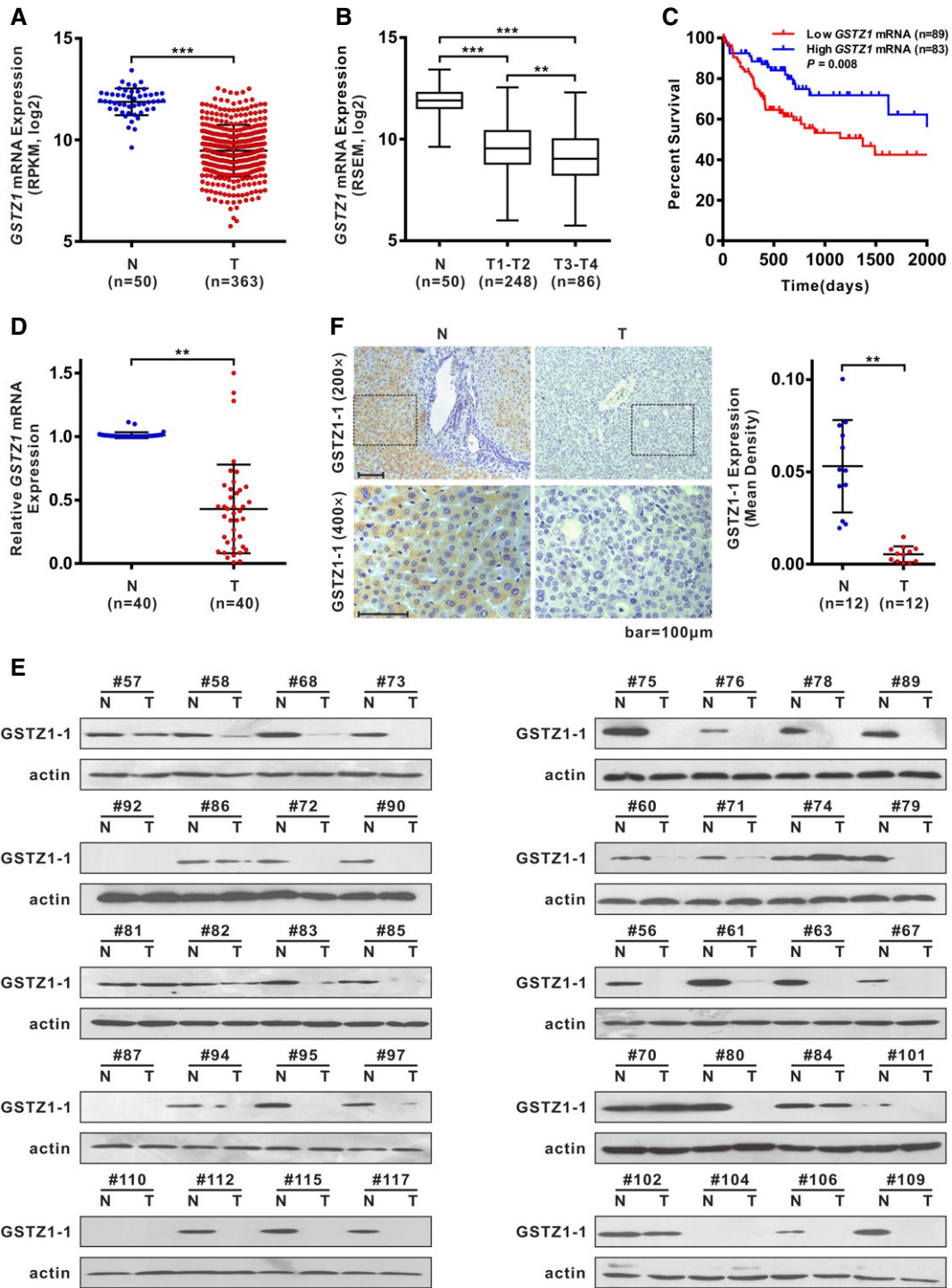

**Figure 1. GSTZ1-1 expression is generally downregulated in human HCC tissues and correlates with poor survival of HCC patients.**

A   *GSTZ1* mRNA expression in HCC and paired non-tumor tissues from The Cancer Genome Atlas (TCGA) Liver Hepatocellular Carcinoma (LIHC) dataset.

B   *GSTZ1* mRNA expression in increasing pathologic stages of HCC based on data from TCGA database. The box-and-whisker plots display the medians (horizontal lines), interquartile ranges (boxes), and minimum and maximum values (whiskers) of the mRNA expression data. The n indicates the number of samples.

C   Overall survival of HCC patients with high (> 25 percentile) or low (≤ 25 percentile) mRNA expression of GSTZ1-1, based on TCGA data.

D, E   *GSTZ1* mRNA (D) and protein (E) expression in 40 cases of HCC and paired non-tumor tissues.

F   Representative images (left) and quantification (right) of GSTZ1-1 immunohistochemical staining in 12 cases of HCC and paired non-tumor tissues. Scale bars: 100 μm.

Data information: **P < 0.01, ***P < 0.001, Student's *t*-test (A, D, and F) or one-way ANOVA followed by the Tukey test (B) or Gehan–Breslow–Wilcoxon test (C).
Abbreviations: N, non-tumor; T, tumor; RPKM, reads per kilobase per million mapped reads; RSEM, RNA sequencing by expectation maximization.

Source data are available online for this figure.

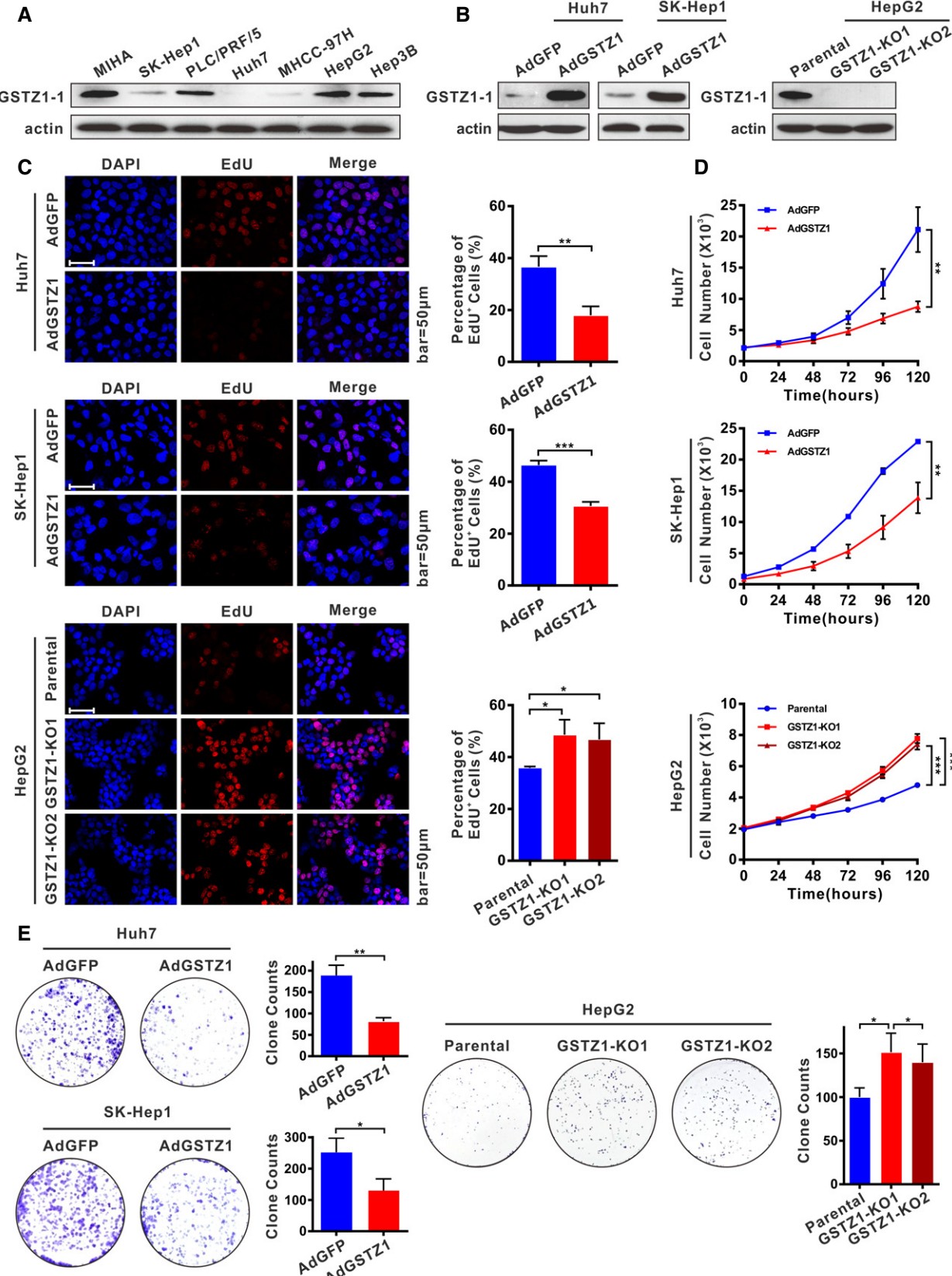

Figure 2.

◀

**Figure 2. GSTZ1-1 inhibits cell proliferation in HCC cell lines.**

A    Western blotting shows endogenous GSTZ1-1 protein expression in HCC cell lines and MIHA cells.
B    Overexpression of GSTZ1-1 in Huh7 (left) and SK-Hep1 (middle) cells and knockout of *GSTZ1* in HepG2 (right) cells were confirmed by immunoblot assay. The GSTZ1-1-overexpressing (GSTZ1-OE) cell model was established by infecting hepatoma cells with adenoviruses expressing GSTZ1-1 (AdGSTZ1). Adenoviruses expressing green fluorescent protein (AdGFP) were used as a control. The GSTZ1 knockout (GSTZ1-KO) cell model was established using the CRISPR/Cas9 system.
C–E  Proliferation ability of GSTZ1-OE Huh7 and SK-Hep1 cells, and GSTZ1-KO HepG2 cells. (C) Representative images (left) and quantification (right) of EdU-positive cells. Scale bars: 50 μm. (D) Cell growth curves. (E) Representative images (left) and quantification (right) of colony formation capacity. Values are shown as means ± SD ($n = 3$), *$P < 0.05$, **$P < 0.01$, ***$P < 0.001$, Student's $t$-test (two groups) or one-way ANOVA followed by the Tukey test (more than two groups). Abbreviations: DAPI, 4′,6-diamidino-2-phenylindole; EdU, 5-ethynyl-2′-deoxyuridine.

Source data are available online for this figure.

observed activation of the NRF2 pathway in $Gstz1^{-/-}$ mouse liver tissues (Fig EV3B). Metabolites of Phe/Tyr catabolism in the murine liver were analyzed by ultra-high-performance liquid chromatography coupled to triple-quadrupole mass spectrometry (UHPLC-QqQ-MS). Prior to liver tissue collection, mice were supplemented with 2% (w/v) Phe in their drinking water for 1 week. UHPLC-QqQ-MS analysis indicated that SA concentrations in $Gstz1^{-/-}$ mouse livers were 4.5-fold higher than those in WT mouse livers (Fig 3D). These results suggested that loss of GSTZ1-1 resulted in SA accumulation and NRF2 activation.

We then checked whether treatment with Phe and SA could activate the NRF2 signaling pathway in HepG2 cells. Based on a luciferase activity, either Phe (2.0 mM) or SA (200 μM) activated ARE-based transcription (Fig 3E). Immunoblotting showed that the expression of nuclear NRF2 protein as well as its downstream target NQO1 was dramatically increased in HepG2 cells in the presence of Phe or SA (Fig 3F). These results indicated that overloading of the Phe/Tyr catabolic pathway activated the KEAP1/NRF2 pathway.

**Succinylacetone stabilizes NRF2 via alkylation of cysteine residues of KEAP1**

As an electrophilic metabolite, SA may have reactive properties similar to those of fumarate and also act as a Michael acceptor to react with intracellular cysteines. We assumed that SA may activate NRF2 by alkylation of KEAP1 cysteine residues, similar to cysteine modification by fumarate (Adam *et al*, 2011) or itaconate (Mills *et al*, 2018). To analyze KEAP1 alkylation directly, GSTZ1-KO HepG2 cells were transfected with Flag-tagged KEAP1 and then treated with Phe (final concentration, 2.0 mM). MS analysis of Flag-immunoprecipitated KEAP1 identified SA modification at residues Cys23, Cys319, Cys406, and Cys513 (Table EV2, Figs 3G, and EV4). Next, we checked Myc-tagged NRF2 (NRF2-Myc) expression in HepG2 cells co-expressing WT KEAP1 or C23S, C151S, C319S, C406S, or C513S mutant KEAP1. SA stabilized NRF2-Myc in HepG2 cells co-expressing WT KEAP1, but not in HepG2 cells co-expressing C151S and C406S mutant KEAP1. Replacing C23, C319, or C513 with serine generated nonfunctional KEAP1 mutants, which could not mediate degradation of NRF2-Myc, even without SA treatment (Fig 3H). These site-directed mutagenesis experiments indicated that C406 and C151 of KEAP1 play a critical role in sensing SA. However, our MS analysis did not identify SA modification at C151. Together, these results demonstrated that accumulation of SA contributed to NRF2 activation via alkylating KEAP1 cysteine residues.

**GSTZ1-1 expression negatively correlates with IGF1R in HCC cell lines and HCC tissues**

The mechanism underlying HCC suppression by GSTZ1-1 was explored by RNA sequencing of AdGSTZ1- and AdGFP-infected Huh7 cells. As illustrated by the volcano plot, GSTZ1-OE Huh7 cells showed that expression of *IGF1R* mRNA was significantly decreased relative to that of AdGFP controls (fold change > 1.5 or < 0.667, FDR < 0.05, $n = 512$; Fig 4A). Further, mRNA and protein expression of IGF1R were downregulated in GSTZ1-OE Huh7 cells but upregulated in GSTZ1-KO HepG2 cells (Fig 4B and C). Moreover, a correlation analysis of 363 patients with HCC included in the TCGA Liver Hepatocellular Carcinoma (LIHC) dataset showed a negative correlation between mRNA expression of GSTZ1-1 and IGF1R ($r = -0.31$, $P < 0.0001$, $n = 363$; Fig 4D left). qRT–PCR, immunoblot, and IHC assays revealed that the expression of GSTZ1-1 was negatively correlated with that of IGF1R in HCC tissues ($r = -0.68$, $P < 0.0001$, $n = 30$; Fig 4D right and Fig 4E–F). Together, these results suggest that GSTZ1-1 expression is negatively related to that of IGF1R in HCC.

**GSTZ1-1 suppresses IGF1R-mediated antiapoptotic pathway in hepatoma cells**

To evaluate the effects of GSTZ1-1 and IGF1R on cell proliferation capacity, we performed growth assays in GSTZ1-KO HepG2 cells treated with or without picropodophyllin (PPP), a specific kinase inhibitor of IGF1R. Functional studies indicated that *GSTZ1* knockout promoted proliferation. The accelerated cell growth in HepG2 cells lacking GSTZ1-1 was considerably abated by PPP in a dose-dependent manner, indicating the essential role of IGF1R in HCC tumorigenesis by GSTZ1-1 depletion (Fig 5A and B). To examine whether the growth-inhibition effect of GSTZ1-1 is associated with apoptosis, we measured apoptosis induction by examining the apoptotic ratio of HCC cells. The flow cytometry assays revealed the percentage of early apoptotic cells remarkably increased ($P < 0.0001$) in GSTZ1-OE Huh7 cells, while it decreased significantly ($P = 0.0065$) in GSTZ1-KO cells. Consistently, PPP treatment induced obvious apoptotic induction in GSTZ1-1-deficient HepG2 cells (Figs 5C and EV5A). These data suggested that tumor-promotion effects by GSTZ1-1 depletion might be related to IGF1R-mediated antiapoptotic pathway.

Our data obtained by RNA sequencing and qRT–PCR indicated that differential expression of IGF1R, and its downstream signaling molecules and apoptosis-associated genes, was consistent with GSTZ1-1-induced apoptotic phenotype (Fig 5D–E). Thus, we

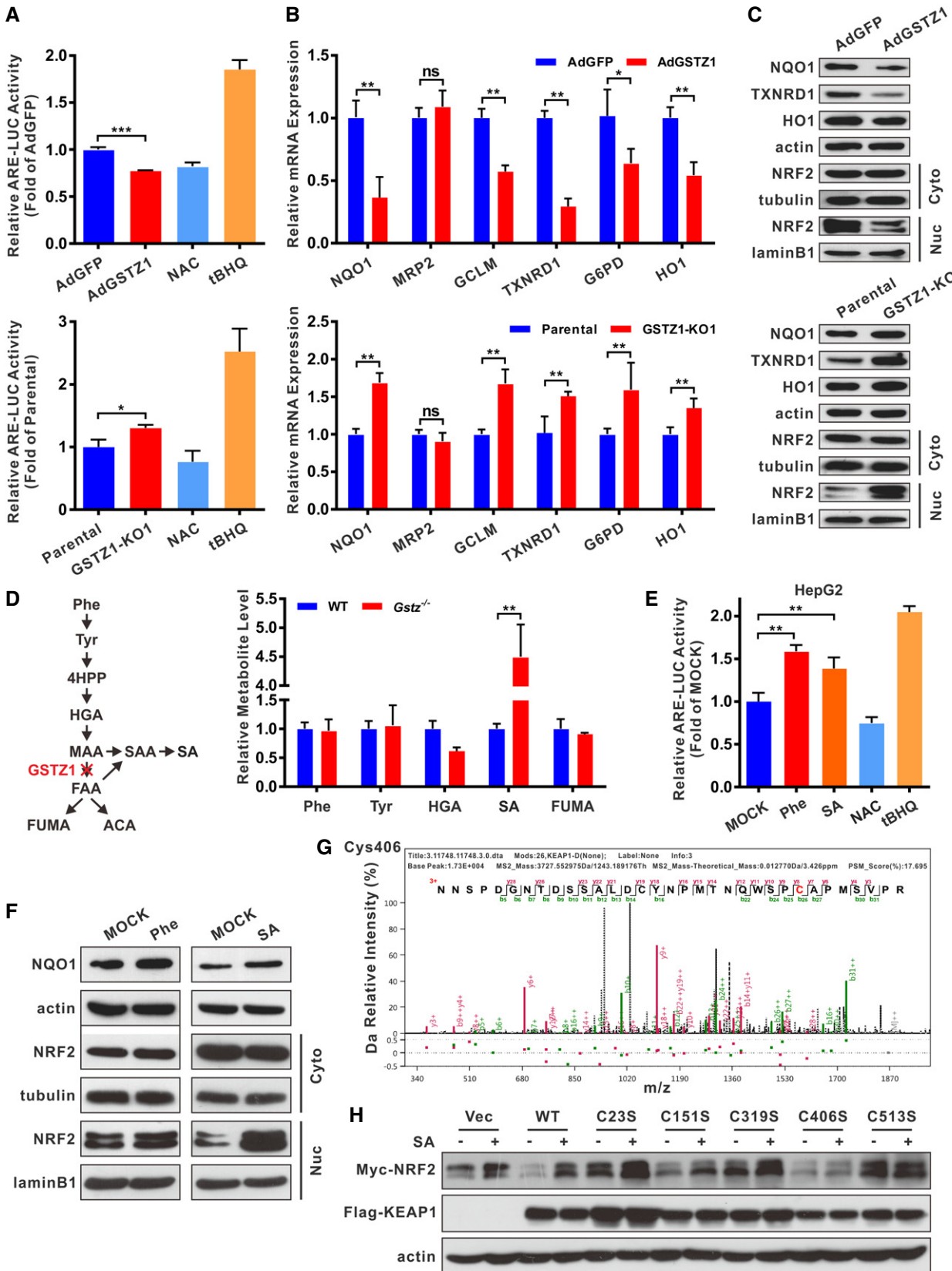

Figure 3.

**Figure 3.** **GSTZ1-1 deficiency activates the KEAP1/NRF2 pathway via succinylacetone accumulation.**

A Luciferase (LUC) activity of ARE promoter in GSTZ1-1-overexpressing (GSTZ1-OE) Huh7 cells (top) and *GSTZ1* knockout (GSTZ1-KO) HepG2 cells (bottom). The antioxidant N-acetylcysteine (NAC, 20 mM for 12 h) and the NRF2 activator tertiary butylhydroquinone (tBHQ, 40 μM for 3 h) were used as negative and positive controls, respectively.

B, C Relative mRNA (B) and protein (C) expression of NRF2 downstream target genes (such as NQO1, TXNRD1, and HO1), and cytoplasmic (Cyto) and nuclear (Nuc) expression of NRF2 (C) in GSTZ1-OE Huh7 cells (top) and GSTZ1-KO HepG2 cells (bottom).

D Phenylalanine and tyrosine (Phe/Tyr) catabolic pathway (left) and relative levels of metabolites in tyrosine catabolism in murine liver as measured by mass spectrometry (right).

E LUC activity of ARE promoter in HepG2 cells treated with Phe (2.0 mM) and succinylacetone (SA, 200 μM) for 36 h. NAC and tBHQ were described as above.

F Western blotting shows NQO1 expression and cytoplasmic and nuclear expression of NRF2 in HepG2 cells treated with Phe (2.0 mM, left) and SA (200 μM, right) for 48 h.

G SA alkylation of human KEAP1 cysteine residues on Cys406 was identified in GSTZ1-KO HepG2 cells infected with adenoviruses expressing KEAP1 (AdKEAP1) by MS/ MS analysis.

H Western blotting shows NRF2 and KEAP1 expression in HepG2 cells cotransfected with NRF2-Myc and wild-type (WT) or mutant KEAP1 (C23S, C151, C319S, C406S, or C513S).

Data information: Values are shown as means ± SD (*n* = 3), *P < 0.05, **P < 0.01, ***P < 0.001, Student's *t*-test (two groups) or one-way ANOVA followed by the Tukey test (more than two groups). Abbreviations: NQO1, NAD(P):H quinone oxidoreductase 1; MRP2, multidrug resistance protein 2; GCLM, glutamate-cysteine ligase modifier subunit; TXNRD1, thioredoxin reductase 1; G6PD, glucose-6-phosphate dehydrogenase; HO1, heme oxygenase 1; ns, not significant; 4HPP, 4-hydroxyphenylpyruvate; HGA, homogentisic acid; MAA, maleylacetoacetate; FAA, fumarylacetoacetate; FUMA, fumarate; ACA, acetoacetate; SAA, succinylacetoacetate; Vec, vector.

Source data are available online for this figure.

detected the expression of IGF1R, and its downstream genes and apoptosis-associated proteins, in gain and loss of GSTZ1-1 hepatoma cells. Previous studies suggested that antiapoptotic effect of IGF1R is mediated by PI3K/AKT and Ras/ERK signaling pathways through the BCL2-associated death protein (BAD) and cAMP-response element binding protein (CREB) (phosphorylated at Ser133; Datta *et al*, 1997; Wang *et al*, 1999). We found that GSTZ1-OE cells exhibited lower phosphorylation levels of AKT, ERK, BAD, and CREB. Moreover, increased cleavage of both cysteine aspartic acid specific protease 9 (caspase 9) and caspase 3, but decreased levels of the antiapoptotic protein B cell lymphoma/leukemia-2 protein (BCL2), was observed in GSTZ1-OE cells (Fig 5F–G). On the contrary, GSTZ1-1 ablation altered AKT, ERK, BAD, CREB phosphorylation, and apoptosis-associated gene expression in an opposite way (Fig 5F–G). To confirm the role of IGF1R in the antiapoptotic effect of GSTZ1-1 deficiency in HCC cells, we pretreated GSTZ1-KO cells with the IGF1R inhibitor PPP or IGF1R shRNA. As expected, treatment with PPP or IGF1R shRNA blocked the antiapoptotic effect of GSTZ1-1 depletion in GSTZ1-1-deficient cells (Fig 5F–G). These results indicate that GSTZ1-1 mediated proapoptotic effects in HCC cells by suppressing the IGF1R signaling pathway.

## Upregulation of IGF1R mediated by GSTZ1-1 deficiency is NRF2-dependent

We then explored the connection between activation of NRF2 and enhanced expression of IGF1R induced by GSTZ1-1 depletion. For this, we analyzed the transcriptional activities and protein levels of IGF1R by regulating NRF2 activation in GSTZ1-KO and GSTZ1-OE HCC cells (Fig 6A–C). *GSTZ1* knockout increased the luciferase activity of human *IGF1R* promoter and protein expression of IGF1R, but this effect was reversed by treatment with brusatol (Bru), an inhibitor of NRF2 (Fig 6A). Similar inhibitory effects on *IGF1R* transcription and expression were observed during overexpression of KEAP1 in GSTZ1-KO cells (Fig 6B). Conversely, GSTZ1-1 overexpression decreased *IGF1R* promoter activity and protein level, whereas treatment with the NRF2 activator tertiary butylhydroquinone (tBHQ) rescued these changes (Fig 6C). These results suggest that GSTZ1-1 affects the expression of IGF1R by regulating NRF2 activity.

Next, we explored the underlying mechanisms of how NRF2 regulates IGF1R expression. NRF2 interacts with specificity protein 1 (SP1) (Gao *et al*, 2014), a major transcription factor of *IGF1R*; therefore, we speculated that NRF2 may regulate *IGF1R* transcription by binding and recruiting SP1 to the *IGF1R* promoter region. To verify this hypothesis, we first assessed the interaction of NRF2 and SP1 proteins using co-immunoprecipitation assays in MHCC-97H cells. The results confirmed the interaction between NRF2 and SP1 (Fig 6D). Subsequently, chromatin immunoprecipitation assays also revealed that inactivation of NRF2 by Bru treatment blocked the recruitment of SP1, whereas NRF2 activator tBHQ increased SP1 binding to the *IGF1R* promoter (Fig 6E). Taken together, these data demonstrate that NRF2 activation, induced by *GSTZ1* knockout, may function as a coactivator and promote *IGF1R* transcription via interaction with SP1.

Furthermore, we examined whether treatment with Phe or SA regulates the expression levels of IGF1R and NRF2 in HCC cell lines. We found that exogenous Phe and SA loading dose-dependently increased the expression of IGF1R and NRF2 in HepG2 cells (Fig 6F). Nuclear translocation of NRF2, induced by treatment with Phe and SA, was confirmed by immunoblotting (Fig 3F). Phe and SA increased the activity of human *IGF1R* promoter and promoted SP1 enrichment on the *IGF1R* promoter (Figs 6G and EV5B). Interestingly, 2-(2-nitro-4-trifluoromethylbenzoyl)-1,3-cyclohexanedione (NTBC), an inhibitor of 4-hydroxyphenylpyruvate dioxygenase, which blocked the production of homogentisate and downstream metabolites, reversed Phe-mediated upregulation of IGF1R. However, it failed to inhibit IGF1R expression induced by SA (Fig 6H). Altogether, these findings suggest that SA accumulation, induced by *GSTZ1* knockout, increased IGF1R expression via NRF2 activation and SP1 enrichment on the *IGF1R* promoter.

## *GSTZ1* knockout promotes DEN/CCl₄-induced hepatocarcinogenesis via activation of IGF1R-mediated antiapoptotic pathway in mice

To further identify how GSTZ1-1 mediates IGF1R in hepatocarcinogenesis *in vivo*, we used the DEN/CCl₄-induced mouse model of liver cancer. Treatment with DEN/CCl₄ was used to induce

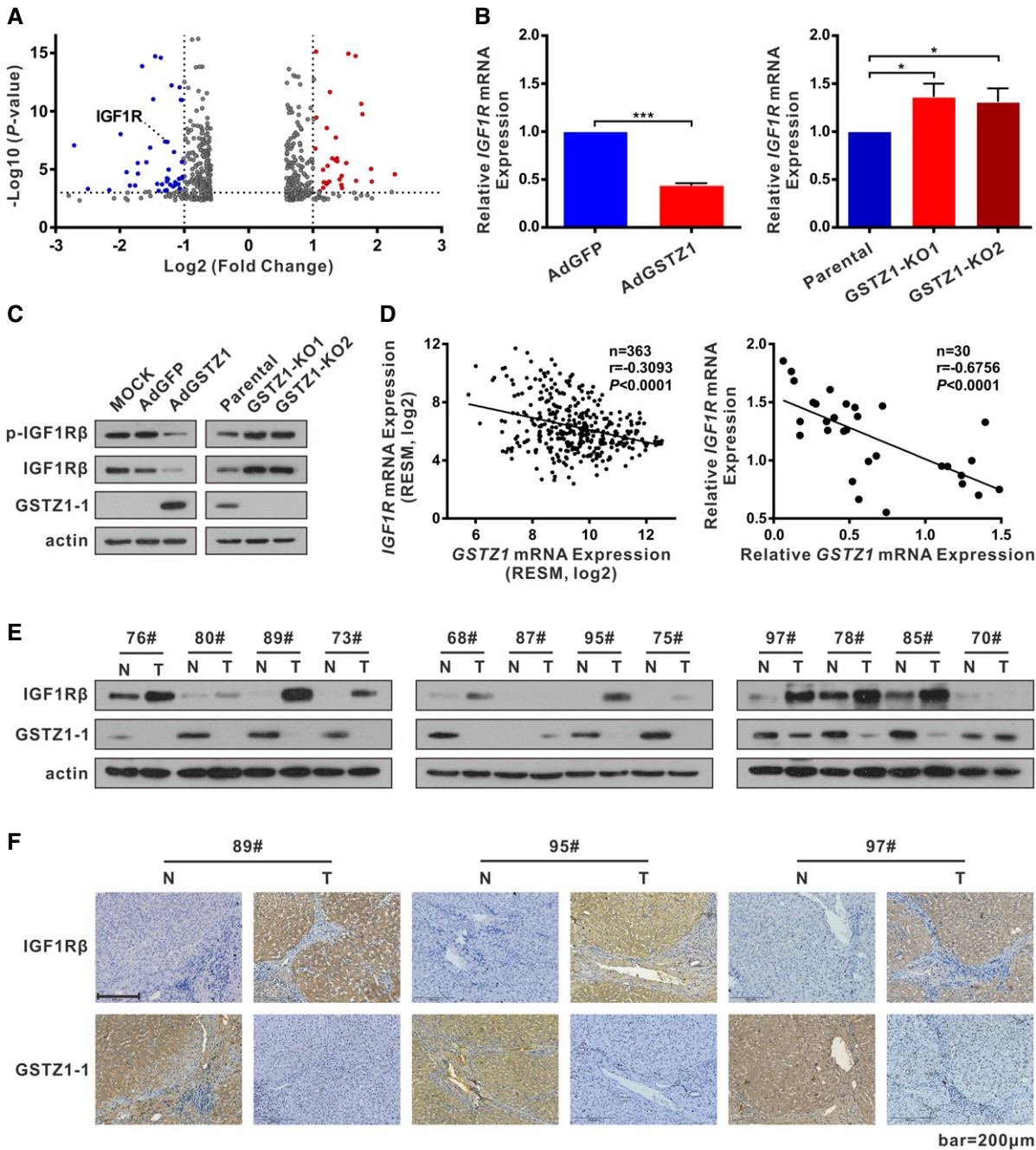

**Figure 4. GSTZ1-1 expression negatively correlates with IGF1R in HCC cell lines and HCC tissues.**

A    Volcano plot of RNA-sequencing data obtained using GSTZ1-1-overexpressing (GSTZ1-OE) Huh7 cells (fold change > 1.5 or < 0.667, FDR < 0.05, n = 512).

B, C    *IGF1R* mRNA (B) and protein (C) expression in GSTZ1-OE Huh7 (left) and *GSTZ1* knockout HepG2 (right) cells.

D    Correlation analysis of *GSTZ1* and *IGF1R* mRNA expression was conducted using data from 363 patients with HCC included in TCGA LIHC dataset (left) and 30 cases of HCC and paired non-tumor tissues (right).

E    Western blotting shows GSTZ1-1 and IGF1Rβ expression in HCC and paired non-tumor tissues (n = 12).

F    Representative images of GSTZ1-1 and corresponding IGF1Rβ immunohistochemical labeling in HCC and paired non-tumor tissues. Scale bars: 200 μm.

Data information: Values are shown as means ± SD (n = 3), *P < 0.05, ***P < 0.001, Student's *t*-test (two groups) or one-way ANOVA followed by the Tukey test (more than two groups) (B) or Pearson r test (D). Abbreviations: p-IGF1Rβ, phospho (Tyr1131)-IGF1Rβ; RSEM, RNA sequencing by expectation maximization; N, non-tumor; T, tumor.

Source data are available online for this figure.

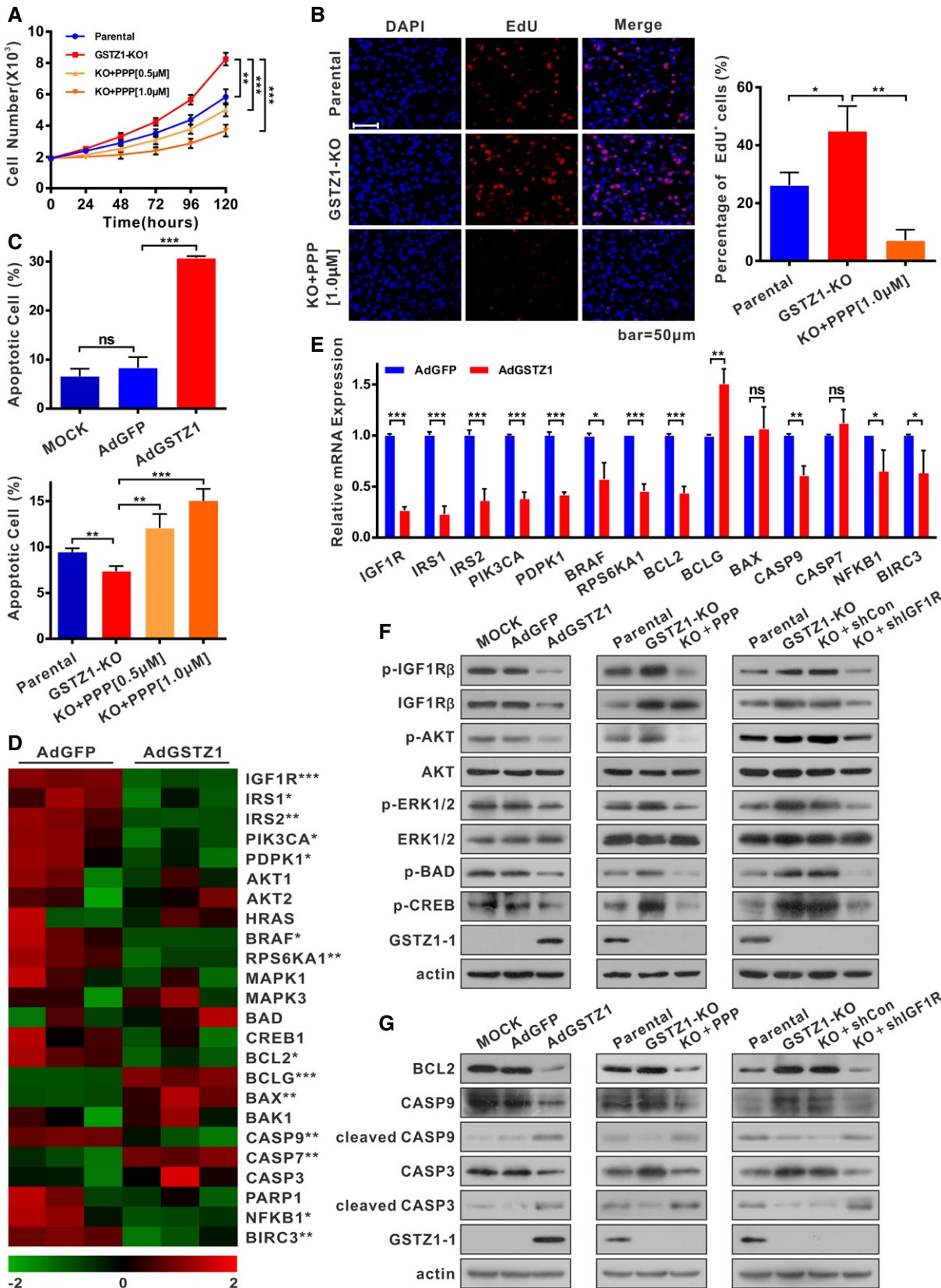

**Figure 5.**

Figure 5. GSTZ1-1 suppresses IGF1R-mediated antiapoptotic pathway in HCC cell lines.

A, B    Proliferation ability of *GSTZ1* knockout (GSTZ1-KO) HepG2 cells treated with or without different doses of picropodophyllin (PPP, 0.5 μM and 1.0 μM) for 24 h. (A) Cell growth curves. (B) Representative images (left) and quantification (right) were obtained by EdU incorporation assays. Scale bars: 50 μm.
C       Flow cytometry detection of apoptosis, conducted using Annexin V–FITC/PI double staining and quantification, shows early apoptotic cells in GSTZ1-OE Huh7 (top) and GSTZ1-KO HepG2 (bottom) cells. GSTZ1-KO cells were treated with PPP as described above.
D, E    Heat map of RNA-sequencing data (described in Fig 4) (D) and quantitative RT–PCR (E) show the differential expression of downstream signaling molecules and apoptosis-associated genes in IGF1R-mediated antiapoptotic pathway. The asterisks indicate statistically significant differences between AdGSTZ1 and AdGFP.
F, G    Expression of IGF1R downstream signaling molecules (F) and apoptosis-associated proteins (G) in GSTZ1-OE Huh7 (left) and GSTZ1-KO HepG2 (middle, right) cells. GSTZ1-KO cells were treated with or without PPP (1.0 μM) for 24 h (middle) and transfected with control (shCon) or IGF1R shRNA (shIGF1R) (right). All cells were treated with (F) or without (G) IGF-1 (Beyotime, 50 ng/ml) for 1 h after serum starvation overnight.

Data information: Values are shown as means ± SD ($n = 3$), *$P < 0.05$, **$P < 0.01$, ***$P < 0.001$, Student's *t*-test (two groups) or one-way ANOVA followed by the Tukey test (more than two groups). Abbreviations: DAPI, 4′,6-diamidino-2-phenylindole; EdU, 5-ethynyl-2′-deoxyuridine; ns, not significant; p-IGF1Rβ, phospho (Tyr1131)-IGF1Rβ; p-AKT, phospho (Thr308)-AKT; p-ERK1/2, phospho (Thr202/Tyr204)-ERK 1/2; p-BAD, phospho (Ser136)-BAD; p-CREB, phospho (Ser133)-CREB; BCL2, B cell lymphoma/leukemia-2 protein; CASP, caspase.

Source data are available online for this figure.

hepatocarcinoma in the mice, and the mice were also subjected to a Phe overloading treatment (0.25% Phe in the drinking water from 3 weeks after birth) to increase the burden on the Phe/Tyr catabolic pathway (Fig 7A). *Gstz1*$^{-/-}$ mice exhibited promoted liver tumorigenesis with increased tumor masses, number of tumor nodules, and higher levels of alanine aminotransferase (ALT) and alpha-fetoprotein (AFP) in serum (Fig 7B and C). The immunoblot assay indicated that protein expression of IGF1R and BCL2 was increased, and the expression of cleaved caspase 3 was decreased, in liver tumors of *Gstz1*$^{-/-}$ mice (Fig 7D). In contrast, mice treated with PPP or Bru exhibited slower tumor growth and decreased number of tumor nodules compared with those of untreated *Gstz1*$^{-/-}$ mice (Fig 7B and C). Furthermore, treatment with PPP or Bru induced robust apoptosis by inducing increased expression of cleaved caspase 3; this was evidenced by the weak signal of phosphorylated IGF1R as assessed by immunoblot and IHC assays (Fig 7D and E). H&E staining indicated that treatment with PPP or Bru alleviated the disordered structure and arrangement of tumor tissue (Fig 7E).

Collectively, these data indicate that GSTZ1-1 depletion increased susceptibility to DEN/CCl$_4$-induced carcinogenesis and promoted hepatocarcinogenesis via IGF1R-mediated antiapoptotic pathway. GSTZ1-1 functions as an important tumor suppressor by inhibiting NRF/IGF1R axis in HCC (Fig 7F). Targeting NRF2 or IGF1R may be a promising treatment approach for this subset HCC.

# Discussion

Several inborn errors of metabolism (IEMs) can predispose humans to cancers related to the accumulation or depletion of metabolites. HCC is one of the most common cancers observed in carriers of IEMs (Erez & DeBerardinis, 2015). Genetic diseases due to deficiencies of all enzymes in Phe/Tyr catabolic pathway have been well documented, with the exception of GSTZ1-1 (Chakrapani *et al*, 2012). Hereditary tyrosinemia type 1 (HT1) is caused by deficiency of the final enzyme in Phe/Tyr catabolism pathway, fumarylacetoacetate hydrolase (FAH), and is characterized by progressive liver diseases with increased risk for HCC (Weinberg *et al*, 1976; Van Spronsen *et al*, 1994). However, to date, clinical data are not sufficient to indicate a role for GSTZ1-1 in inherited genetic disorders. Recently, six children with mild hypersuccinylacetonemia caused by sequence variants in *GSTZ1* were reported; however, no evidence of liver dysfunction was detected (Yang *et al*, 2017).

The current study revealed that GSTZ1-1 expression is markedly downregulated in HCC, which contributed to tumor progression and poor prognosis, providing evidence supporting recent reports that GSTZ1-1 is decreased in HCC (Jahn *et al*, 2016; Nwosu *et al*, 2017). Moreover, our studies demonstrated that GSTZ1-1 plays a tumor suppressor role in HCC. We suggest that loss of GSTZ1-1 results in the accumulation of the oncometabolite SA, alkylation of KEAP1, and NRF2 activation, promoting *IGF1R* transcription by recruiting

Figure 6. Upregulation of IGF1R mediated by GSTZ1-1 deficiency is NRF2-dependent.

A–C    Luciferase (LUC) activity of the human *IGF1R* promoter (left) and expression of NRF2 and IGF1Rβ (right) in *GSTZ1* knockout (GSTZ1-KO) HepG2 and GSTZ1-1-overexpressing (GSTZ1-OE) Huh7 cells. GSTZ1-KO cells were treated with or without brusatol (Bru, 40 nM) for 24 h (A) and transfected with pSEB-Vector or pSEB-Keap1 (B) GSTZ1-OE cells were treated with or without tertiary butylhydroquinone (tBHQ, 40 μM) for 3 h (C).
D       Co-immunoprecipitation assay shows NRF2–SP1 interactions in MHCC-97H cells.
E       Chromatin immunoprecipitation assay was conducted using extracts of MHCC-97H cells treated with or without Bru (40 nM) for 24 h (left), and with or without tBHQ (40 μM) for 3 h (right). IgG and anti-Histone H3 (H3) were used as negative and positive controls, respectively.
F       Expression of IGF1Rβ and NRF2 in HepG2 cells treated with phenylalanine (Phe, 0, 0.5, 1.0, 2.0, and 4.0 mM, top) and succinylacetone (SA, 0, 50, 100, 200, and 500 μM, bottom) for 48 h, as assessed by Western blotting.
G       Chromatin immunoprecipitation assay of extracts from MHCC-97H cells treated with or without Phe (2.0 mM, left) and SA (200 μM, right) for 48 h; IgG and anti-Histone H3 were described as above.
H       IGF1Rβ expression in HepG2 cells treated with Phe (2.0 mM, top) and SA (200 μM, bottom) for 48 h, then respectively added 2-(2-nitro-4-trifluoromethylbenzoyl)-1,3-cyclohexanedione (NTBC, 0, 4, 7, 14, 28 μg/ml) for the last 12 h.

Data information: Values are shown as means ± SD ($n = 3$), *$P < 0.05$, **$P < 0.01$, ***$P < 0.001$, Student's *t*-test (two groups) or one-way ANOVA followed by the Tukey test (more than two groups). Abbreviations: Vec, vector; ns, not significant.

Source data are available online for this figure.

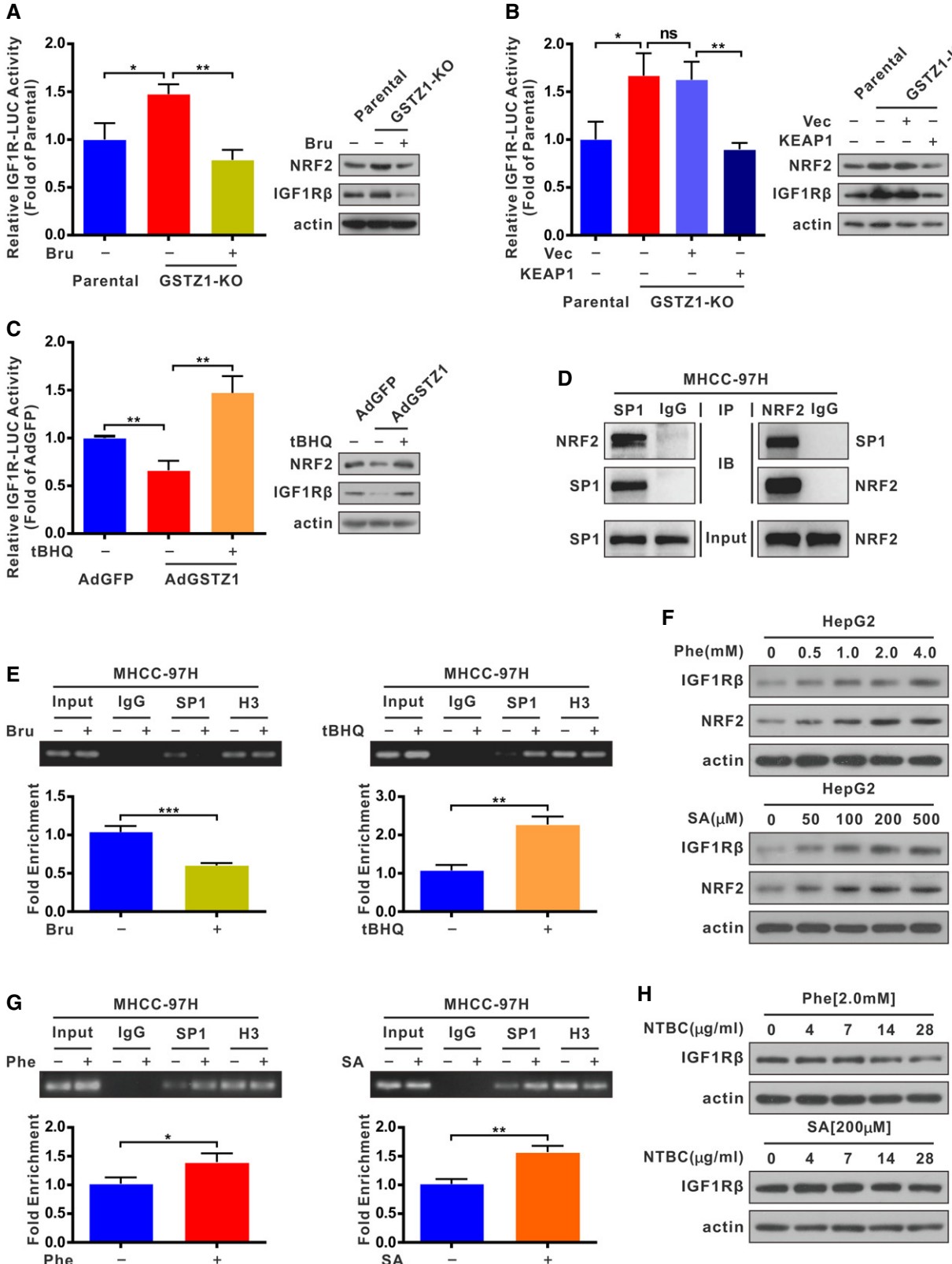

Figure 6.

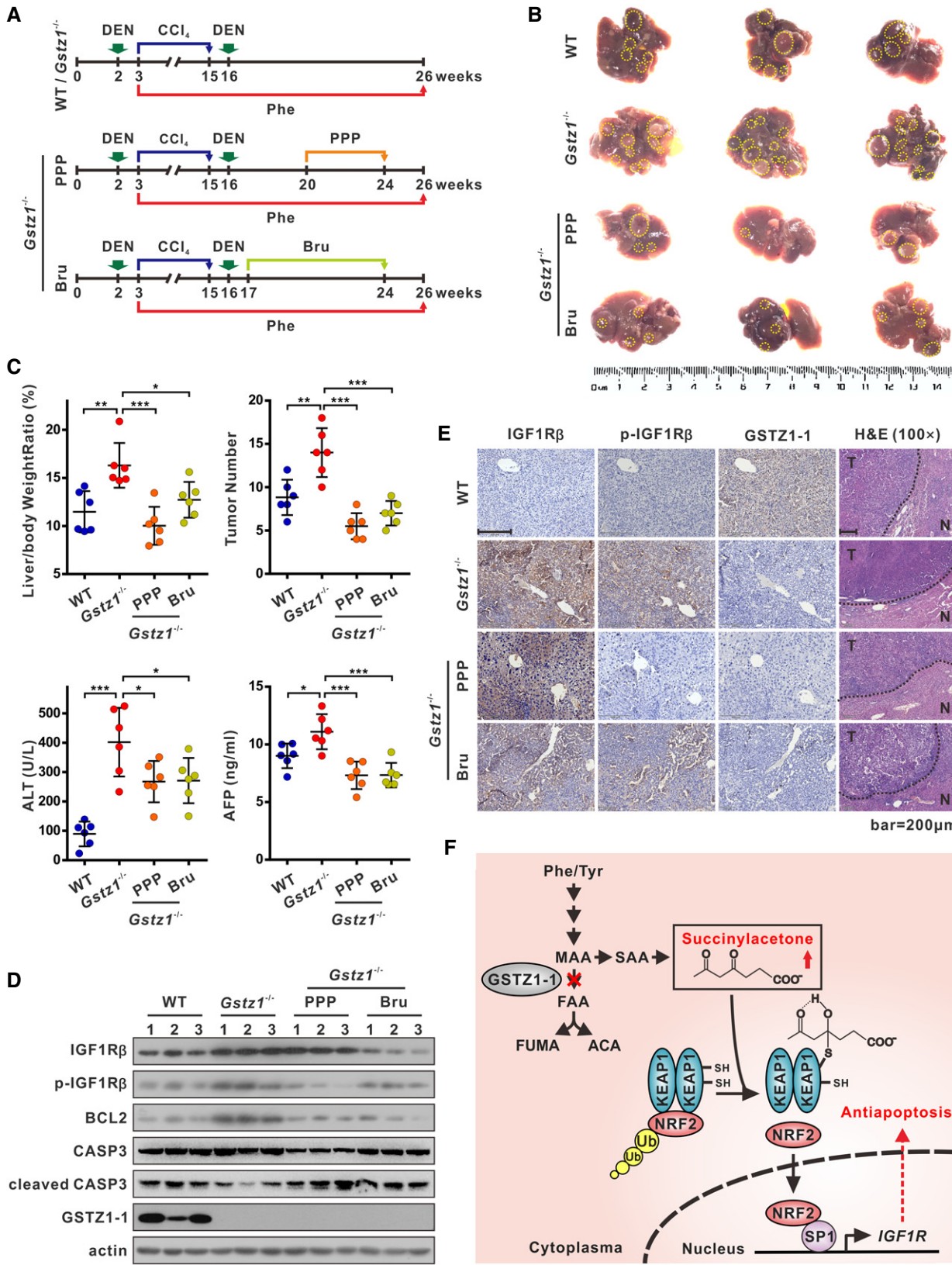

Figure 7.

**Figure 7.  *GSTZ1* knockout promotes HCC tumorigenesis *in vivo* via activation of IGF1R-mediated antiapoptotic pathway.**

A  Schematics showing experimental design for procedures involving wild-type (WT) and *Gstz1*$^{-/-}$ mice.

B  Gross appearances of liver with tumors. The yellow dotted-line circles represent tumors.

C  Liver/body weight ratio (top left), tumor numbers (top right), and serum alanine aminotransferase (ALT) (bottom left) and alpha-fetoprotein (AFP) (bottom right) levels of the four groups.

D  Protein expression of IGF1Rβ, phospho (Tyr1131)-IGF1Rβ, B cell lymphoma/leukemia-2 protein (BCL2), and caspase 3 (CASP3) in four groups of liver tumors as assessed by Western blotting.

E  Representative images of GSTZ1-1, and corresponding IGF1Rβ and phospho (Tyr1161)-IGF1Rβ immunohistochemical labeling, and H&E staining in liver tumors. Scale bars: 200 μm.

F  A proposed model of activation of the NRF2/IGF1R axis in HCC.

Data information: Values are shown as means ± SD obtained using six mice in each group, *$P < 0.05$, **$P < 0.01$, ***$P < 0.001$, one-way ANOVA followed by the Tukey test**.** Abbreviations: DEN, diethylnitrosamine; CCl$_4$, carbon tetrachloride; Phe, phenylalanine; PPP, picropodophyllin; Bru, brusatol; H&E, hematoxylin–eosin; N, non-tumor; T, tumor; Tyr, tyrosine; MAA, maleylacetoacetate; FAA, fumarylacetoacetate; FUMA, fumarate; ACA, acetoacetate; SAA, succinylacetoacetate; SA, succinylacetone; Ub, ubiquitination.

Source data are available online for this figure.

SP1 to its promoter, which promotes the antiapoptotic pathway of hepatoma cells *in vitro* and *in vivo*.

KEAP1 is a key negative regulator of NRF2 ubiquitination and proteasomal degradation. Mutation, deletion, oxidation, or alkylation of KEAP1 allows NRF2 to accumulate, migrate into the nucleus, and activate the expression of target genes (Hayes & Dinkova-Kostova, 2014; Suzuki & Yamamoto, 2015). Several MS studies have identified endogenous metabolites that directly modified the KEAP1 cysteine residues. Fumarate hydratase inactivation results in fumarate accumulation, succination of the KEAP1 cysteine residues, and NRF2 activation in renal carcinomas (Adam *et al*, 2011). The endogenous metabolite itaconate has also been shown to activate NRF2 by alkylation of KEAP1 cysteine residues (Mills *et al*, 2018). Our study revealed a novel post-translational modification of KEAP1: SA directly alkylates cysteine residues 23, 319, 406, and 513. KEAP1 is a cysteine-rich protein that acts as a sensor for different electrophiles. Our site-directed mutagenesis experiments indicated that C406 and C151 of KEAP1 play a critical role in sensing SA.

Previous studies demonstrated that FAA and SA accumulated in the urine of *Gstz1*$^{-/-}$ 129SvJ mice (Fernández-Cañón *et al*, 2002), and SA accumulated in the serum of *Gstz1*$^{-/-}$ BALB/c mice (Lim *et al*, 2004). We detected SA accumulation in the liver tissues of *Gstz1*$^{-/-}$ 129SvJ mice by targeted MS; however, we did not detect MAA, FAA, or succinylacetoacetate (SAA) as they are highly reactive and unstable. SA is thought to be a stable derivative of MAA and FAA, and elevated blood level of SA is currently a primary screening marker to detect HT1 in newborns (De Jesús *et al*, 2014). Interestingly, NRF2 activation was also observed in *FAH*$^{-/-}$ mice (Marhenke *et al*, 2008), but the mechanism underlying NRF2 activation was not assessed. Sandhu predicted that FAA activates NRF2 by covalently modifying cysteine residues of KEAP1 (Sandhu *et al*, 2015). Because SA accumulated in HT1 as well, our MS/MS data suggested that SA may also contribute to NRF2 activation in *FAH*$^{-/-}$ mice via direct modification of KEAP1. However, because of the lack of specific antibody, we were unable to quantify KEAP1 modification on cysteine residues by SA in GSTZ1-KO cells. Whether metabolites MAA and FAA also accumulate and modify KEAP1 in this model requires further study. Based on the existing literature (Erez & DeBerardinis, 2015) and our findings, SA can be described as an oncometabolite that might promote malignant cellular transformation and progression of HCC.

Interestingly, DCA, a substrate and mechanism-based inactivator of GSTZ1-1 (Tzeng *et al*, 2000), also reverses the Warburg effect in cancer cells and has been proposed for use in targeted therapy against cancer (Michelakis *et al*, 2008). However, as previously reported by the International Agency for Research on Cancer (IARC, 2014), DCA induces liver cancer in both mice and rats (DeAngelo *et al*, 1999; Bull *et al*, 2002). Although the evidence suggesting carcinogenicity of DCA in humans is inadequate, several studies indicate that DCA exposure could lead to hepatic toxicity (Ammini & Stacpoole, 2003). According to our study, DCA may release the oncogenic potential of SA by inhibiting GSTZ1-1. Given that in cancer therapy DCA is likely to be given in combination with cytotoxic drugs, future studies should consider the potential tumourigenicity of such combinations.

The expression of IGF1R is determined, to a large extent, at the transcriptional level. The typical NRF2 binding motif (5′-TGACnnnGC-3′) was not found in the *IGF1R* promoter region. Our results show that *IGF1R* promoter activities and protein levels were positively correlated with NRF2 activity, suggesting that NRF2 indirectly affected the transcription of *IGF1R*. SP1 is a ubiquitous zinc-finger transcription factor that binds with high affinity to the GC-rich motif at multiple promoters. SP1 is a potent transactivator of the *IGF1R* gene, whereas several tumor suppressors, including breast cancer gene 1 (BRCA1), P53, and Wilms' tumor protein 1 (WT1), repress the activity of SP1 (Werner & Sarfstein, 2014). Previous studies have shown that NRF2 can interact with SP1 and c-Jun for binding and transcriptionally regulating several downstream genes (Gao *et al*, 2014). In agreement with these findings, our results show that NRF2 interacted with SP1 and regulated SP1 enrichment on the *IGF1R* promoter. Moreover, Phe overloading and exogenous SA also promoted the binding of SP1 to the *IGF1R* promoter and increased transcription of IGF1R. Therefore, SA accumulation in *Gstz1*$^{-/-}$ mice may be an initial factor that activates NRF2, induces IGF1R expression, and subsequently promotes the progression of HCC. Furthermore, the efficiency of PPP treatment in our mouse model provides evidence supporting that targeting IGF1R is beneficial for HCC therapy.

Brusatol treatment led to somewhat decreased expression of cytoplasmic NRF2 and distinctly decreased expression of nuclear NRF2 and IGF1R in *Gstz1*$^{-/-}$ mice. Brusatol enhances NRF2 degradation in several cell lines, including A549 and Hepa-1c1c7 cells, and sensitizes them to antitumor drugs (Zhu *et al*, 2016). We infer that Bru inhibits the NRF2/IGF1R axis in *Gstz1*$^{-/-}$ mice, inducing

apoptosis of tumor cells. High constitutive activation of NRF2 has been found in carcinomas of the lungs, liver, gallbladder, ovary, breast, and stomach (Sporn & Liby, 2012). Our results indicate that NRF2 may serve as a potential therapeutic target for HCC. However, in a mouse model of HT1, NRF2 activation seemed to protect $Fah^{-/-}$ mice from FAA-induced liver injury and delay hepatocarcinogenesis (Marhenke *et al*, 2008). It is currently commonly accepted that NRF2 plays dual roles in protecting or promoting tumorigenesis at different stages of tumorigenesis (Rojo de la Vega *et al*, 2018). Considering the dual roles of NRF2 in cancer (Lau *et al*, 2008), the specific role of NRF2 activation in DEN/CCl$_4$-induced hepatocarcinogenesis requires further study using, for example, *Gstz1* and *Nrf2* double knockout mice and investigating its function in different stages of tumorigenesis.

In summary, our results suggested that GSTZ1-1 is downregulated in HCC and may serve as a prognostic marker. Our results indicate an alternative oncogenic action of SA through activation of the NRF2/IGF1R axis by alkylation of KEAP1. The findings provide a novel mechanistic link between disrupted metabolism and tumorigenesis and indicate the potential of targeting NRF2/IGF1R axis for personalized therapy in HCC.

# Materials and Methods

### Patient samples

Human HCC tissues and paired non-tumor tissues were obtained from patients who underwent surgery for HCC at the First and the Second Affiliated Hospital of Chongqing Medical University. Sampling was conducted with the approval of the Research Ethics Committee of Chongqing Medical University, and informed consent was obtained from participating patients. Each sample was frozen immediately after surgery and stored in liquid nitrogen for later use.

### $Gstz1^{-/-}$ mouse study

Heterozygous 129-$Gstz1^{\mathrm{tm1Jmfc}}$/Cnbc mice (EM: 04481) were purchased from the European Mouse Mutant Archive via Nanjing Biomedical Research Institute of Nanjing University and were crossed to breed wild-type (WT) and $Gstz1^{-/-}$ mice. All mice were maintained under specific pathogen-free conditions in the laboratory animal center of Chongqing Medical University. The mice were divided into four groups: WT group, $Gstz1^{-/-}$ group, $Gstz1^{-/-}$ + picropodophyllin (PPP; MCE, Monmouth, NJ, USA) group, and $Gstz1^{-/-}$ + brusatol (Bru; Meilunbio, Dalian, Liaoning, China) group. Each group included three male and three female mice. At 2 weeks of age, all mice were administered an intraperitoneal injection of diethylnitrosamine (DEN; Sigma, St. Louis, MO, USA) at a dose of 75 mg/kg. At week 3, the mice were administered 0.25% (w/v) Phe in drinking water until sacrifice and were injected with carbon tetrachloride (CCl$_4$; Macklin, Shanghai, China) intraperitoneally at a dose of 2 ml/kg twice a week for 12 weeks. At 16 weeks of age, the mice were injected intraperitoneally with another dose of DEN (50 mg/kg). In the $Gstz1^{-/-}$ + PPP group at 20 weeks, the $Gstz1^{-/-}$ mice were administered an intraperitoneal injection of PPP at a dose of 10 mg/kg every 12 h for 4 weeks. In the $Gstz1^{-/-}$ + Bru group at 17 weeks, the $Gstz1^{-/-}$ mice were

administered an intraperitoneal injection of Bru at a dose of 2 mg/kg every 2 days for 7 weeks. Body weight was measured and retro-orbital blood collection was carried out before sacrifice. The four groups of mice were sacrificed at 26 weeks of age. Liver weight with tumors was measured, and the number of tumors was counted. Liver tumor proteins were extracted for Western blotting. Samples of liver tumor tissue, with or without adjacent normal tissue, were fixed in 4% paraformaldehyde, embedded in paraffin, and sectioned for hematoxylin–eosin (H&E) staining or immunohistochemical assay. All animal procedures were performed according to protocols approved by the Rules for Animal Experiments published by the Chinese government. All procedures were also approved by the Research Ethics Committee of Chongqing Medical University (reference number: 2017010).

### Statistical analysis

Statistical analysis and data plotting were performed using GraphPad Prism 7. Data are presented as the means ± SD. Unless mentioned otherwise, Student's t-test was used to compare between two groups, and one-way ANOVA followed by the Tukey test was used to compare among more than two groups. Statistical significance was defined as $P < 0.05$.

For detailed descriptions of other methods, please refer to Appendix Supplementary Methods.

# Data availability

RNA-Seq data: Gene Expression Omnibus GSE117822 (https://www.ncbi.nlm.nih.gov/geo/query/acc.cgi?acc=GSE117822).

Expanded View for this article is available online.

### Acknowledgements

We would like to thank Dr. T.-C He (University of Chicago, USA) for providing the plasmid pAdEasy system. We are grateful to Prof. Ding Xue (School of Life Sciences, Tsinghua University) for supplying the CRISPR/Cas9 system, lenti-CRISPR v2, pMD2.G, and psPAX2. We also thank Prof. Yiguo Zhang for providing the pGL3-ARE plasmid. This work was supported by the China National Natural Science Foundation (Grant Nos. 81872270 and 81572683 to N.T.; 81661148057 to A.L.H.; and 81602417 to K.W.), the Major National S&T program (2017ZX10202203-004 to N.T.), Natural Science Foundation Project of CQ CSTC (cstc2018jcyjAX0254 to N.T.), the Program for Innovation Team of Higher Education in Chongqing (CXTDX201601015), the Leading Talent Program of CQ CSTC (CSTCCXLJRC201719 to N.T.), the Scientific Research Innovation Project for Postgraduate in Chongqing (CYB17119), and Talent Development Program of CQMU for Postgraduate (BJRC201704).

### Author contributions

NT, AH, and KW conceived the study and designed the experiments. FY also designed part of the experiments. FY, JL, and HD performed most experiments and analyzed the data. YW, LC, and PX performed mass spectrometry analysis. CL and QW assisted with IGF1R pathway analysis. JXiang and JXia assisted with experiments in GSTZ1-1-deficient mice. XP assisted with xenograft assays. LL generated KEAP1 mutants. XL collected clinical HCC samples. QL and XL provided guidance and advice. FY, JL, KW, and NT prepared the manuscript with all authors providing feedback.

## Conflict of interest

The authors declare that they have no conflict of interest.

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
