## [Review Process File · The EMBO Journal]

GSTZ1-1 Deficiency Activates NRF2/IGF1R Axis in HCC via Accumulation of Oncometabolite Succinylacetone

Fan Yang, Jingjing Li, Haijun Deng, Yihao Wang, Chong Lei, Qiuji Wang, Jin Xiang, Li Liang, Jie Xia, Xuanming Pan, Xiaosong Li, Quanxin Long, Lei Chang, Ping Xu, Ailong Huang, Kai Wang, Ni Tang

Review timeline:

Submission date:	7 March 2019
Editorial Decision:	9 April 2019
Revision received:	24 May 2019
Accepted:	2 June 2019

Editor: Daniel Klimmeck

Transaction Report:

1st Editorial Decision

9 April 2019

Thank you for the submission of your manuscript (EMBOJ-2019-101964) to The EMBO Journal. Your manuscript has been sent to two referees, and we have received reports from both of them, which I enclose below.

As you will see, the referees acknowledge the potential high interest and quality of your work, and they are supportive of publication of your manuscript in The EMBO Journal, pending minor revision. In more detail, referee #2 requests clarification of GSTZ1's contribution to basal conditions. Further, referee #1 asks you to complement the discussion considering the GSTZ1 substrate DCA and its usage in cancer therapies.

I judge the comments of the referees to be generally reasonable and given their interest, we are happy to invite you to revise your manuscript to address the referees' comments.

REFeree REPORTS

Referee #1:

GSTZ1-1 plays an essential role in the tyrosine and phenylalanine catabolism pathway. This work shows that in the absence of GSTZ1-1 the intermediate succinylacetone (SA) accumulates and triggers signalling through KEAP1/NRF2/IGF-1R to promote tumour growth. While it has previously been shown that GSTZ1-1 deficiency stimulates an antioxidant response through NRF2 this work significantly extends the earlier studies with more mechanistic detail. In addition this work reveals the involvement of IGF-1R and the tumour suppressor role of GSTZ1-1. This latter discovery is novel and of interest as it demonstrates that SA is one of a few known oncometabolites. The paper is well written and the experimentation appears sound.

There is one interesting deficiency in the discussion. It is recognized by the authors that genetically determined GSTZ1-1 deficiency is rare and it is unlikely that it would be a major cause of HCC.

However dichloroacetic acid (DCA) is a substrate and mechanism based inhibitor of GSTZ1-1 (see Tzeng et al Chem. Res. Toxicol. 2000, 13, 231-236). DCA also reverses the Warburg effect in cancer cells and there is a growing literature advocating its use as a cancer treatment. Consequently if DCA inhibits GSTZ1-1 it may release the oncogenic potential of SA. Given that in cancer therapy DCA is likely to be given in combination with cytotoxic drugs it may be wise for future studies to consider the potential tumourigenicity of such combinations.

A minor point. The proper nomenclature for the protein is GSTZ1-1 and the mouse gene is *Gstz1* in italics. See Mannervik et al METHODS IN ENZYMOLOGY, VOL. 401 pp1-8 2005

Referee #2:

The manuscript by Yang et al. stems from the intriguing observation that the GSTZ1 enzyme of phenylalanine catabolism is down-regulated in liver cancer. The authors propose that Succinylacetone accumulation in GSTZ1 mutant tissues leads to KEAP1 alkylation, NRF2 activation and expression of IGF1R.

The experimental approach is sound and the great quality of the data supports the author conclusions. All the immunoblot analysis, proliferation/survival assays, Mass spectrometry and cancer biopsies data are convincing and properly quantified. The findings are relevant for cancer research, spanning from mechanistic studies, metabolomics and validation in in vivo cancer models. While additional experiments could be added (as always), this study is complete as is and will be of very broad interest. I would recommend clarifying one single issue:

1) The measurements of KEAP1 alkylation (Fig. 3G) and tumor formation (Fig. 7B) are in "stressed" conditions, either after phenylalanine treatment or DEN exposure. However, GSTZ1 modulation has dramatic effects on cell proliferation and survival in basal conditions (Fig. 2). How do the authors explain these effects? What are the Succinylacetone levels in basal conditions in vitro and in vivo? What are the rates of cell proliferation/survival in GSTZ1 mutant livers prior to DEN treatment?

1st Revision - authors' response

24 May 2019

REVIEWER 1

GSTZ1-1 plays an essential role in the tyrosine and phenylalanine catabolism pathway. This work shows that in the absence of GSTZ1-1 the intermediate succinylacetone (SA) accumulates and triggers signalling through KEAP1/NRF2/IGF-1R to promote tumour growth. While it has previously been shown that GSTZ1-1 deficiency stimulates an antioxidant response through NRF2 this work significantly extends the earlier studies with more mechanistic detail. In addition this work reveals the involvement of IGF-1R and the tumour suppressor role of GSTZ1-1. This latter discovery is novel and of interest as it demonstrates that SA is one of a few known oncometabolites. The paper is well written and the experimentation appears sound.

There is one interesting deficiency in the discussion. It is recognized by the authors that genetically determined GSTZ1-1 deficiency is rare and it is unlikely that it would be a major cause of HCC. However dichloroacetic acid (DCA) is a substrate and mechanism based inhibitor of GSTZ1-1 (see Tzeng et al Chem. Res. Toxicol. 2000, 13, 231-236). DCA also reverses the Warburg effect in cancer cells and there is a growing literature advocating its use as a cancer treatment. Consequently if DCA inhibits GSTZ1-1 it may release the oncogenic potential of SA. Given that in cancer therapy DCA is likely to be given in combination with cytotoxic drugs it may be wise for future studies to consider the potential tumourigenicity of such combinations.

Response: We thank the reviewer for the positive and constructive comments. We completely agree that in the future, studies should consider the potential role of dichloroacetic acid (DCA) in tumourigenicity. As reported previously by the International Agency for Research on Cancer (IARC), DCA induces liver cancer in both mice and rats, although the evidence suggesting carcinogenicity of DCA in humans is inadequate (IARC, 2014). Based on the reviewer's suggestion, we have now discussed this in the revised version of our manuscript (on page 17) as follows:

“Interestingly, DCA, a substrate and mechanism-based inactivator of GSTZ1-1 (Tzeng et al, 2000), also reverses the Warburg effect in cancer cells and has been proposed for use in targeted therapy against cancer (Michelakis et al, 2008). However, as previously reported by the International Agency for Research on Cancer (IARC, 2014), DCA induces liver cancer in both mice and rats (DeAngelo et al, 1999; Bull et al, 2002). Although the evidence suggesting carcinogenicity of DCA in humans is inadequate, several studies indicate that DCA exposure could lead to hepatic toxicity (Ammini & Stacpoole, 2003). According to our study, DCA may release the oncogenic potential of SA by inhibiting GSTZ1-1. Given that in cancer therapy DCA is likely to be given in combination with cytotoxic drugs, future studies should consider the potential tumourigenicity of such combinations.”

A minor point. The proper nomenclature for the protein is GSTZ1-1 and the mouse gene is Gstz1 in italics. See Mannervik et al METHODS IN ENZYMOLOGY, VOL. 401 pp1-8 2005.

Response: We thank the reviewer for pointing this out. We have made the necessary changes at all relevant instances.

REVIEWER 2

The manuscript by Yang et al. stems from the intriguing observation that the GSTZ1 enzyme of phenylalanine catabolism is down-regulated in liver cancer. The authors propose that Succinylacetone accumulation in GSTZ1 mutant tissues leads to KEAP1 alkylation, NRF2 activation and expression of IGF1R.

The experimental approach is sound and the great quality of the data supports the author conclusions. All the immunoblot analysis, proliferation/survival assays, Mass spectrometry and cancer biopsies data are convincing and properly quantified. The findings are relevant for cancer research, spanning from mechanistic studies, metabolomics and validation in vivo cancer models.

Response: We thank the reviewer for the positive comments.

While additional experiments could be added (as always), this study is complete as is and will be of very broad interest. I would recommend clarifying one single issue:

1) The measurements of KEAP1 alkylation (Fig. 3G) and tumor formation (Fig. 7B) are in "stressed" conditions, either after phenylalanine treatment or DEN exposure. However, GSTZ1 modulation has dramatic effects on cell proliferation and survival in basal conditions (Fig. 2). How do the authors explain these effects?

Response: We agree and thank the reviewer for this comment. In Figure 2, we illustrated the *in vitro* experiments performed under basal conditions to show the effects of GSTZ1 modulation on hepatoma cell proliferation and survival. In Figure 3G, we illustrated that Flag-tagged KEAP1 was transfected to GSTZ1-KO HepG2 cells to measure its alkylation.

First, as highly efficient enrichment is essential for mass spectrometry (MS) analysis of post-translational modification and since specific antibody for this novel modification is not available commercially, we intended to increase the alkylation stoichiometry of KEAP1 proteins and the sensitivity of detection approaches under “stressed” conditions in GSTZ1-KO cells.

Second, the concentration of intermediate metabolites such as succinylacetone is very low in HepG2 cells in basal conditions (please see also point 2), so we treated the cells with 2.0 mM phenylalanine (Phe) to increase the burden on the Phe catabolic pathway (such as succinylacetone accumulation). In another similar study of KEAP1 alkylation, the authors also detected the modification of KEAP1 by MS under "stressed" condition. They treated cells with 4-octyl itaconate, an itaconate derivative to exacerbate the accumulation of intracellular itaconate before analyzing KEAP1 using tandem MS (Mills *et al*, Nature, 2018). Similarly, HEK293T cells were treated with exogenous methylglyoxal (MGO, 5mM) to determine the MGO modification of KEAP1 by LC-MS/MS analyses (Bollong *et al*, Nature, 2018).

Third, as shown in Figure 7B, *Gstz1*^{-/-} mice were treated with DEN/CCl₄ to induce hepatocellular carcinoma (HCC), as previous studies reported that *Gstz1*^{-/-} mice exhibited focal hepatitis and hepatocyte necrosis, but showed no signs of HCC (Cindy *et al*, Am. J. Pathol., 2004; Fernández-Cañón *et al*, Mol. Cell. Biol., 2002). Diethylnitrosamine (DEN) is a carcinogen that has been previously used to induce HCC in mice (Nat Rev Gastroenterol Hepatol 2018; 15(9):536-554). In our study, the mice were also subjected to a Phe overloading treatment (0.25% Phe in the drinking water) in addition to DEN to examine the roles of GSTZ1 deficiency in HCC tumorigenesis.

Lastly, our *in vitro* functional studies were performed in HCC cell lines (Fig. 2), while *in vivo* experiments were performed in mice (Fig. 7B). The cell types used in the above two experiments were different, and we focused on the role of GSTZ1 in HCC tumorigenesis and progression. So we treated mice with DEN/CCl₄ to induce HCC and added 0.25% Phe to their drinking water to induce stress and consequently increase the burden on the Phe catabolic pathway and exacerbate the accumulation of succinylacetone.

2) What are the Succinylacetone levels in basal conditions *in vitro* and *in vivo*?

Response: Thank you for this valuable comment. We attempted to determine the concentration of succinylacetone (SA) in GSTZ1-KO HepG2 cells by mass spectrometry, since we had hypothesized that endogenous SA in HepG2 cells may increase after *GSTZ1* knockout. However, the levels of SA are probably much lower in HepG2 cells, which is possibly why we failed to detect SA in these cells even after Phe overloading treatment. Additionally, SA detection in cell models was not reported as well. Since we opted not to show the original data in the paper, we have added the data here for the perusal of the reviewer (Rebuttal Fig. 1). We apologize that we were unable to determine the SA levels *in vitro* owing to technical limitations.

As requested, we detected SA levels in the liver tissues of wild-type (WT) and *Gstz1*^{-/-} mice under basal conditions (without Phe and DEN treatment) by UHPLC-QqQ-MS. The results showed that the SA concentrations in the liver of *Gstz1*^{-/-} mice was 4.07 nmol/g, whereas there was no detectable SA in the liver of WT mice (please see Rebuttal Fig. 2). These results were consistent with previously published data which stated that SA accumulated in the urine and serum of *Gstz1*^{-/-} mice, as estimated by GC-MS or its capacity to inhibit δ -aminolevulinic acid dehydratase (Fernández-Cañón *et al*, Mol. Cell. Biol., 2002; Cindy *et al*, Am. J. Pathol., 2004).

3) What are the rates of cell proliferation/survival in *GSTZ1* mutant livers prior to DEN treatment?

Response: We appreciate the reviewer's comment. In response to this question, we performed Ki67 proliferation assay and TUNEL apoptosis assay on the liver of WT and *Gstz1*^{-/-} mice (2 weeks) without DEN treatment. Ki67-positive or TUNEL-positive cells were quantified respectively, however, no significant difference was found between WT and *Gstz1*^{-/-} mice (please see Rebuttal Fig. 3). According to the results, the rates of cell proliferation/survival in *Gstz1*^{-/-} mouse livers are consistent with WT mouse livers at the age of 14 days prior to DEN treatment.

Rebuttal Figure 1. Succinylacetone was not detected in GSTZ1-KO HepG2 cells by mass spectrometry. (A) UHPLC-QqQ-MS analysis and (B) UHPLC-MS/MS analysis of succinylacetone in standard solutions (top) and GSTZ1-KO HepG2 cell samples (treated with 2.0 mM phenylalanine) (bottom).

Rebuttal Figure 2. Succinylacetone concentrations in the liver tissues of wild-type (WT) and *Gstz1*^{-/-} mice under basal conditions. Data are presented as mean \pm SD (n=6).

Rebuttal Figure 3. Cell proliferation and survival in wild-type (WT) and *Gstz1*^{-/-} mice livers prior to DEN treatment. (A) Ki67 proliferation assay. (B) TUNEL apoptosis assay. Quantification of Ki67-positive or TUNEL-positive cells are shown. Livers of WT and KO mice were isolated at 14 days of age without DEN treatment. The number of mice is 3 for each group. Data are expressed as mean \pm SD. *ns*, no statistical significance.

2nd Editorial Decision

enter date

Thank you for submitting the revised version of your manuscript. I have now evaluated your amended manuscript and also asked referee #2 to re-assess your revised study (please see his/her comment enclosed below). In light of all these information, I have now concluded that the remaining minor concerns have been sufficiently addressed.

Thus, I am pleased to inform you that your manuscript has been accepted for publication in the EMBO Journal.

REFeree REPORT

Referee #2:

The authors addressed my minor issues.

Corresponding Author Name: Ailong Huang, Kai Wang, Ni Tang

Manuscript Number: EMBOJ-2019-101964